# Robust Reconstruction of the Void Fraction from Noisy Magnetic Flux Density Using Invertible Neural Networks

**DOI:** 10.3390/s24041213

**Published:** 2024-02-14

**Authors:** Nishant Kumar, Lukas Krause, Thomas Wondrak, Sven Eckert, Kerstin Eckert, Stefan Gumhold

**Affiliations:** 1Institute of Software and Multimedia Technology, Technische Universität Dresden, 01187 Dresden, Germany; stefan.gumhold@tu-dresden.de; 2Institute of Process Engineering and Environmental Technology, Technische Universität Dresden, 01069 Dresden, Germany; l.krause@hzdr.de (L.K.); k.eckert@hzdr.de (K.E.); 3Institute of Fluid Dynamics, Helmholtz-Zentrum Dresden-Rossendorf, 01328 Dresden, Germany; t.wondrak@hzdr.de (T.W.); s.eckert@hzdr.de (S.E.)

**Keywords:** machine learning, invertible neural networks, normalizing flows, water electrolysis, Biot–Savart law, inverse problems, current tomography, random error diffusion

## Abstract

Electrolysis stands as a pivotal method for environmentally sustainable hydrogen production. However, the formation of gas bubbles during the electrolysis process poses significant challenges by impeding the electrochemical reactions, diminishing cell efficiency, and dramatically increasing energy consumption. Furthermore, the inherent difficulty in detecting these bubbles arises from the non-transparency of the wall of electrolysis cells. Additionally, these gas bubbles induce alterations in the conductivity of the electrolyte, leading to corresponding fluctuations in the magnetic flux density outside of the electrolysis cell, which can be measured by externally placed magnetic sensors. By solving the inverse problem of the Biot–Savart Law, we can estimate the conductivity distribution as well as the void fraction within the cell. In this work, we study different approaches to solve the inverse problem including Invertible Neural Networks (INNs) and Tikhonov regularization. Our experiments demonstrate that INNs are much more robust to solving the inverse problem than Tikhonov regularization when the level of noise in the magnetic flux density measurements is not known or changes over space and time.

## 1. Introduction

The surging demand for clean energy has led to extensive research into electrolysis as a viable method for greenhouse gas-free hydrogen production [1]. Harnessing excess renewable energy from sources like wind and sunlight enables us to power electrolysis that generates clean hydrogen gas. This hydrogen serves as a reliable energy reservoir, particularly during periods of limited renewable energy availability, thereby addressing the seasonal supply and demand gaps. Moreover, hydrogen exhibits benefits, including extended storage capabilities, presenting a promising solution for reducing carbon footprints [2]. Hydrogen also finds diverse applications, ranging from usage as cryogenic liquid fuel and as a replacement for lithium batteries. However, the overall efficiency of electrolysis faces limitations due to the formation of gas bubbles which block electrodes’ reaction sites and obstruct electric currents [3] as shown in Figure 1. Furthermore, the growth and detachment of bubbles are intricately governed by a complex interplay of forces, including buoyancy, hydrodynamic, and electrostatic forces [4,5,6]. Consequently, detecting both bubble sizes and the location of possible maldistribution of the gas fraction, along with the ability to control bubble formation is critical for ensuring the efficiency and sustainability of hydrogen production through electrolysis.

Detecting bubbles within electrolysis cells is a challenging problem, primarily due to the non-transparency of the electrolyzer structures. A viable and non-invasive solution involves utilizing externally positioned magnetic sensors to capture the bubble-induced fluctuations. However, the availability of only low-resolution magnetic flux density measurements outside the cell, coupled with the high-resolution current distribution inside the cell, necessary to provide accurate bubble information, creates an ill-posed inverse problem for precise bubble detection. To further add to the challenge, the measurement errors originating from sensor noise amplify the difficulty associated with bubble detection.

Contactless Inductive Flow Tomography (CIFT), introduced by Stefani et al. [7], stands as a pioneering method for reconstructing flow fields within conducting fluids, an ill-posed linear inverse problem. This technique leverages Tikhonov regularization to estimate the fluid motion from the measured flow-induced magnetic field under the influence of an applied magnetic field. The data for this reconstruction are obtained from magnetic sensors strategically positioned on the external walls of the fluid volume. However, the reconstruction of the conductivity distribution is an ill-posed non-linear inverse problem that does not induce current through an external magnetic field. Moreover, linear models, such as Tikhonov regularization, demonstrate high sensitivity to noise, particularly when there exists a significant disparity in the amplitude of noise between the data used for model fitting and testing. Also, the limited number of available sensors compounds the difficulty in achieving a satisfactory reconstruction of the high-dimensional current distribution.

Advanced Machine Learning (ML) techniques such as Deep Neural Networks (DNNs) offer a data-driven approach for reconstructing the current distribution within an electrolysis cell. By leveraging external magnetic flux density measurements, these techniques are capable of capturing relationships between the measured magnetic flux density and the internal current distribution of the cell. A method known as Network Tikhonov (NETT) [8] combines DNNs with Tikhonov regularization, where the regularization weightage parameter plays a crucial role in balancing data fidelity and regularization terms. However, the choice of the weightage parameter is based on some heuristic assumptions [9].

Given the limitations of the conventional approaches, we explored the feasibility of Invertible Neural Networks (INNs) to solve our ill-posed non-linear inverse problem. It was recently shown by Ardizzone et al. [10] that INNs are a good candidate for solving such tasks. INNs are marked by a bijective mapping and inherent invertibility between input and output spaces, which present a pragmatic solution for addressing the complexities in estimating the conductivity from relatively much lower resolution of magnetic flux density measurements. Therefore, we studied its performance in comparison to the Tikhonov regularization to estimate the binary conductivity distribution. The binary conductivity represents the non-conducting void fraction as zeros, indicating the presence of bubbles. A cluster of zeros can indicate either the existence of large bubbles or a cluster of small bubbles, enabling us to estimate the void fraction. Our key contributions are:We introduce a novel method that uses INNs to reconstruct the spatial distribution of the void fraction from limited magnetic flux density measurements, thereby addressing the inverse problem of the Biot–Savart Equation in electrolysis.We show that INN is more accurate than the Tikhonov approach to reconstruct the distribution of the void fraction when the amplitude of the noise in the magnetic sensor measurements is not known or varies considerably in space and time.In scenarios where the number of sensors is further reduced, and the distance of the sensor placement from the region where the conductivity needs to be reconstructed is further increased, we show that our INN model is able to provide a good reconstruction of the void fraction distribution.We present a new evaluation metric named random error diffusion that computes the likelihood that the predicted conductivity distribution resembles the ground truth. Based on random error diffusion, we show that our INN-based approach is better than the Tikhonov regularization.

In Section 2, we review the related work,  Section 3 details our simulation setup that mimics electrolysis, while Section 4 elaborates on our INN model and random error diffusion metric. Section 5 presents experimental results, while Section 6 summarizes our main contributions, and discusses the broader application of INNs in process tomography.

## 2. Related Work

This section presents an overview of the related works and is structured into four sub-sections. Section 2.1 delves into the works that discuss the bubble formation as a significant obstacle to efficient hydrogen production. Section 2.2 explores methods that provide analytical solutions for addressing the ill-posed inverse problem in process tomography, including setups that deal with the Biot–Savart Law. Furthermore, Section 2.3 presents a review of conventional deep learning approaches for solving inverse problems, while Section 2.4 examines works that utilize INNs for tackling inverse problems.

### 2.1. Electrolysis for Clean Hydrogen: Notable Challenges

A recent study [11] discusses the challenge posed by the supply–demand mismatch in renewable energy sources such as solar and wind power to achieve a stable and sustainable energy grid. Another related work [12] explores the impact of fluctuations in energy production due to weather conditions and variables like climate change, emphasizing periods of excess energy or insufficient supply that can affect grid stability. Hydrogen production through electrolysis emerges as a promising solution to this issue, utilizing excess renewable energy during periods of abundance to power the electrolysis process. This allows for the generation and storage of hydrogen, which can then be converted back into electricity or used directly in various applications when the renewable energy supply is low [13]. Serving as an energy reservoir, hydrogen production through electrolysis effectively bridges the gap between fluctuating renewable energy production and consistent demand. Additionally, hydrogen’s versatility as a clean fuel makes it a valuable resource for transportation and chemical industry, thereby reducing dependence on fossil fuels and mitigating environmental impacts [13]. Consequently, hydrogen production through electrolysis represents a key strategy for achieving a reliable and sustainable energy system [13].

However, the formation of bubbles poses a significant challenge in the process of electrolysis. As an electrochemical reaction occurs at the electrodes, gas bubbles—typically hydrogen and oxygen—are generated. These bubbles represent the desired product in many electrolytic processes, but they can also impede the efficiency of the reaction [3,14]. The accumulation of bubbles around the electrodes can obstruct the active sites, leading to increased resistance within the electrolysis cell [3,14]. This resistance necessitates higher energy input to sustain the desired current flow. Additionally, if left unmanaged, excessive bubble formation can result in operational issues and reduced efficiency [3,14]. Therefore, understanding and effectively managing bubble dynamics is crucial for optimizing the performance of electrolysis and ensuring the economical production of hydrogen.

Hence, bubble detection in electrolysis plays a critical role in optimizing the efficiency of the process. However, it is a challenging endeavor due to the complex dynamics within the electrolysis cell, and the non-transparent walls of the cell make direct visual observation impractical [15,16]. Instead, researchers often resort to indirect methods, such as utilizing magnetic sensors to detect the magnetic field disturbances caused by the movement of bubbles. These sensors are strategically placed outside the cell to minimize interference and provide reliable tracking of bubble behavior. Upon applying cell voltage to the electrolyzer, an electric current starts to flow. Consequently, this current induces a magnetic field in the vicinity of the electrolytic cell, governed by the Biot–Savart law. Therefore, such a setup may help in designing a more precise and efficient electrolysis system, which should ultimately contribute to advancements in clean and sustainable energy production.

### 2.2. Solving Inverse Problem of Biot–Savart Equation—Analytical Approaches

To the best of our knowledge, no prior research has addressed inverse problems within an electrolysis cell setup. However, works such as [17,18] have focused on solving inverse problems in the context of fuel cells. Wieser et al. [17] introduced a contactless magnetic loop array for estimating current distribution within fuel cells, while [18] designed a magnetic field analyzer with sensors associated with a ferromagnetic circuit that enhanced magnetic field variations, leading to a more precise analysis of the current distribution in fuel cells. The work by Roth et al. [19] proposed to reconstruct a 2D current distribution using Fourier analysis in order to better interpret the magnetometer signals that may be useful in applications like in geophysical surveys. Similarly, [20] investigated the possibility of using magneto–optic imaging to directly observe current distributions in thin superconducting samples. Hauer et al. [21] presented magnetotomography, a non-invasive method to visualize the fuel cell current distribution by measuring magnetic flux with a 3D magnetic sensor and a four-axis positioning system. This method, enabled the precise calculation of current flow within the cell since there was no feedback effect. In the application of plasma physics, work such as [22] introduced Bayesian modeling for inferring the current distribution from measurements of magnetic field and flux, where the plasma current is represented as a grid of toroidal current-carrying solid beams with rectangular cross sections.

### 2.3. Solving Inverse Problems Using Deep Learning

With the advancement in machine learning algorithms, many deep learning approaches have been proposed to tackle inverse problems in medical imaging, including computed tomography [8,23] and magnetic resonance imaging [24]. Works such as [23] proposed a partially learned method by integrating prior information of the ill-posed inverse problem of 2D tomography with a data-driven trainable neural network, while [25] explored deep image prior techniques in the context of ill-posed inverse problems. The work by [24] advocates for Convolutional Neural Networks (CNNs) as the choice for solving the inverse problem of medical image reconstruction and regularizing the network with a deep learned noise prior. Whereas [8] suggests using a neural network named Network Tikhonov (NETT) in conjunction with a Tikhonov regularizer to solve the inverse problem for medical imaging. Similarly, iNETT [26] is another recent method that combines Tikhonov regularization with neural networks, differing from [8] in that the non-stationarily iterated Tikhonov method avoids exhaustive tuning of the regularization parameter. Reference  [27] developed a method for the fast convergence of neural networks used for solving inverse problems in imaging by reducing latency in calculating gradients. To explore more related works dealing with solving inverse problems in medical imaging or imaging in general via deep neural networks, readers are referred to [28,29,30,31,32]. Recent works such as [33] highlight that Deep Neural Networks (DNNs) trained to solve inverse problems are robust to noise and adversarial perturbations. Nevertheless, we believe that fine-tuning the regularization weightage when DNNs are trained with some regularization strategy is challenging, even though methods such as [34] learn such regularization weights.

Machine learning-based approaches have been proposed to solve ill-posed inverse problems in Electrical Capacitance Tomography (ECT) [35,36], Electrical Impedance Tomography (EIT) [37,38], Electrical Resistance Tomography (ERT) [39,40,41], positron emission tomography [42], X-ray tomography [43,44], and novel applications such as electromagnetic inverse scattering using microwaves [45,46], generally via CNNs. A work by [47] explored the reason why CNNs are a good candidate for solving specific inverse problems, where they showed that the usage of convolution framelets represents the input data by convolving local and global information, aiding in learning underlying features in the data. Although CNNs show promise in solving inverse problems, their inherent non-invertibility may undermine their reliability. Other works to solve inverse problems via deep learning, especially adversarial networks [48,49,50] and LSTM-based autoencoder [51], face challenges in ensuring stable training due to their high complexity, making them less suitable for a wide variety of inverse problems.

 Based on our survey on solving inverse problems via deep learning, we conclude that while significant progress has been made in developing such data-driven models, open questions persist regarding invertibility during training, scalability, and reliability of these deep learning-based approaches in applications of process tomography. Therefore, there is a need to explore novel network architectures and address challenges for the wider practical deployment of such machine learning models in scientific domains.

### 2.4. Invertible Neural Networks (INNs)

INNs are a promising new category of deep learning architectures that are inherently invertible in nature. Recently, Ardizzone et al. [10] showed the effectiveness of INNs for solving the inverse problem of predicting the level of oxygenation in tissues from endoscopic images. Even though there have been recent attempts to use INNs as surrogate models for solving inverse problems, such as [52] for inverse problems in physical systems governed by Partial Differential Equations (PDEs), Ref. [53] for inverse problem in morphology, Ref. [54] for inverse problem in medical imaging, or [55] for inverse design of optical lenses. However, INNs remain largely unexplored in the field of solving inverse problems in process tomography. INNs are popularly implemented based on Normalizing Flows (NFlows) that are suitable generative models due to their invertible architectural design, and accurate density estimation [56]. Additionally NFlows do not suffer from posterior collapse, which is common in other generative models such as Variational Auto-Encoders (VAEs) and Generative Adversarial Networks (GANs). NFlows were popularized by [57] for density estimation. Since then, multiple novel NFlows have been proposed in the literature, such as RealNVP [58], Glow [59], FFJORD [60], NAF [61], SOS [62], Cubic Spline Flows [63], and Neural Spline Flows [64]. Each of these prior works differs on the design of the NFlows that includes the design of the coupling function.

In summary, the section showcases the under-explored potential of INNs for addressing the inverse problem of the Biot–Savart Equation and other applications in the industrial process tomography domain in general.

## 3. Simulation Setup

The simulation setup mimics generic features of a water electrolyzer in a simplified model, as depicted in Figure 2 (top). In Section 3.1, we elaborate on the intricate design details related to the simulation. Moving to Section 3.2, we provide information on essential simulation parameters used for the experiment. Subsequently, in Section 3.3, we discuss the mesh transformation step to obtain the fine-grained mesh of the conductivity maps, which will be used as the input to the INN and other evaluated models. In Section 3.4, we formulate the forward physical process of the simulation based on the Biot–Savart Equation and finally, in Section 3.5, we give an overview of the data used to perform the experiments.

### 3.1. Simulation Design

The goal of our simulation setup, depicted in Figure 2 (top), is to investigate the feasibility of localizing and quantifying non-conducting bubbles by reconstructing the conductivity distribution from the observed induced magnetic flux density in the surrounding external region. To achieve this, the simulation setup simplifies the water electrolyzer to a quasi-two-dimensional configuration. The setup is filled with liquid GaInSn as a substitute for water to avoid electrochemical reactions and the generation of additional bubbles. To represent non-conducting gas bubbles, Poly-Methyl Methacrylate (PMMA) cylinders with varying radii and locations are placed throughout the liquid. Hence, the setup incorporates materials with significant conductivity differences to simulate conducting water and low-conducting bubbles. We selected the dimensionality of the simulation setup based on the future experimental setup. The liquid channel’s configuration measures 16×7×0.5 cm. The two Cu electrodes (each measuring 10×7×0.5 cm) facilitate the application of the electric current. The anode and cathode connections to an external power supply are established via wires, modeled with lengths of 50 cm and square cross-sections measuring 0.5 cm on each side.

### 3.2. Simulation Parameters

To compute training data, diverse geometrical setups featuring regions of varying conductivity were compiled from a Java-class file in the finite element software COMSOL Multiphysics V6.0 (COMSOL Inc., Burlington, VT, USA) [65]. This involves placing between 30 and 120 PMMA cylinders with radii ranging from 2 to 2.5 mm within the liquid metal. The cylinder sizes are aligned with bubble agglomerates, and larger clusters are represented by merged cylinders. Since no electrochemical reactions occur in the liquid metal after the application of electric current, concentration-induced conductivity gradients are excluded. A low electrical conductivity of $5\times 10^{-14}$ S/m is employed to simulate the void fraction at PMMA cylinder positions [66]. For the Cu wires and electrodes, values of $5.8\times 10^{7}$ S/m are used, while the liquid metal is assigned a conductivity of $3.3\times 10^{6}$ S/m [67]. A current density of 1 $A/{\mathrm{cm}}^{2}$ is applied at the electrode surface interfacing with the liquid metal, which falls within the typical range for alkaline and PEM electrolyzers. As the input current is conducted through the smaller cross-section copper wire, this necessitates an application of 14 $A/{\mathrm{cm}}^{2}$, corresponding to a total current of 3.5 A.

### 3.3. Mesh Transformation

To facilitate automated grid generation for various bubble distributions, the geometry was discretized using finite tetrahedral elements, forming an unstructured mesh. Following a study to ensure grid independence, the mesh underwent refinement in regions exhibiting high current density gradients, notably at the interfaces between the wire and electrode, as well as within the volume containing liquid GaInSn.  For the liquid metal, the tetrahedral element size of 0.1 mm was set as the minimum, while the maximum was established at 5 mm. The computation of the current and the conductivity distribution for multiple geometries necessitates meshes with varying cell counts. As the INN and other evaluated models require fixed input array dimensions, the initial tetrahedral mesh is transformed into a grid of hexahedrons with a constant number of elements. The current density distribution within the structured mesh, consisting of one cell layer in height, can be treated as two-dimensional, given the negligible influence of the *z*-component and variations in the *x* and *y* components along the *z*-direction of the current. This grid comprises a total of 774 cells, with higher resolution allocated to the middle containing the liquid metal volume, comprising 510 nearly cubic cells, each with dimensions of 4.71×4.67×5 mm. The current density and electrical conductivity within each hexahedron are determined through inverse distance-weighted interpolation [68] utilizing the 24 nearest tetrahedrons.

### 3.4. Solving Forward Process via Biot–Savart Equation

The current distribution $j(r^{\prime })$ was simulated using COMSOL for each bubble distribution, and the magnetic field B(r) exclusively at the positions of virtual sensors, was determined by the Biot–Savart law given as,
(1)$$B\left(r\right)=\frac{\mu_{0}}{4\pi }\int_{V}\frac{j\left(r^{\prime }\right)\times \left(r-r\prime \right)}{{\left|r-r^{\prime }\right|}^{3}}dV$$
where $\mu_{0}$ is the permeability of free space, i.e., a vacuum, given as $4\pi \times 10^{7}$ $N/A^{2}$, *V* is the volume with dV as an infinitesimal volume element and $B\left(r\right)\;\in R^{3}$ is the magnetic flux density at point r with $r^{\prime }$ as the integration variable and a location in *V*. Since only one spatial component of B(r) will be measurable in the planned experimental validation setup, we aim to reconstruct the conductivity distribution by using one spatial component of B(r) that is most informative about the magnetic flux density. Therefore, we selected the *x*-component of the magnetic flux density. The simulation of the current distribution typically requires 2.5 min. Additionally, the mesh transformation, along with calculating the magnetic field using Equation (Equation 1), requires around 3.5 min. Note that the inverse process reconstruction with our INN model typically completes in less than 1 s.

### 3.5. Simulation Data

 To measure the magnetic flux density B(r), we positioned an array of 10×10 virtual sensors, i.e., M=100, at a distance *d* below the liquid GaInSn.  In our future experimental setup, only one spatial component of the magnetic flux density, i.e., the *x*-component is measurable. Thus, the conductivity distribution σ(r′) and one spatial component of the magnetic flux density B(r) serve as the ground truth for every geometrical configuration. We simulated the conductivity distribution for 10,000 different geometrical configurations with a fixed applied current strength of 3.5 A. After transforming the tetrahedral mesh into a hexahedral mesh with fixed dimensions, the resulting conductivities were divided by $\sigma_{GaInSn}=3.3\times 10^{-6}$ S/m, yielding relative conductivities $\sigma_{rel}$ between 0 and 1. Subsequently, $\sigma_{rel}$ were binarized by assigning values smaller than 0.25 as 0 and others as 1. Two examples of binary conductivity maps are shown in Figure 2 (bottom). We selected only those conductivity points directly above the sensor positions. Hence, out of the original 774 simulated conductivity data points, only 510 data points were chosen for each simulated geometry. For each of the 10,000 configurations, the magnetic flux density was calculated at a distance d=5 and 25 mm for 50 and 100 sensor array (see Section 3.2).

## 4. Method

In this section, we provide details related to the INN model and present the developed metrics to evaluate the performance of the model. The section is organized into four main sub-sections. In Section 4.1, we delve into the architecture of the proposed INN framework for addressing the inverse problem of the Biot–Savart Equation. Additionally, Section 4.2 provides a detailed discussion of the loss function employed for training the INN. Following this, in Section 4.3, we elucidate our random error diffusion metric, which helps in assessing the quality of the conductivity reconstruction. To evaluate the robustness of the INN for reconstructing conductivity distribution when there is noise in sensor readings, Section 4.4 presents our algorithm for computing the per-pixel bias and deviation maps.

### 4.1. INN Architecture

 Let us reformulate the conductivity distribution $\sigma \left(r^{\prime }\right)$ as variable x at discretized locations and the strongest spatial component of induced magnetic flux density $B\left(r\right)$ as variable y at distinct locations below the liquid metal. The setup for training the INN, as shown in Figure 3, closely follows Ardizzone et al. [10]. Given that the conductivity map x is an *N*-dimensional vector such that $x\in R^{N}$ and the magnetic flux density measurements y is *M*-dimensional such that $y\in R^{M}$ where N>M, the transformation x→y is non-bijective and thus information loss occurs. We formulate an additional latent variable as $z\in R^{N-M}$ such that for the INN shown in Figure 3, the dimensionality of [y,z] is equal to the dimensionality of x. It is to be noted that the conductivity distribution x, the induced magnetic flux density y and the latent dimension z do not represent the Cartesian xyz coordinates of three-dimensional space of the simulation setup in Figure 2.

The proposed INN model *f* is a series of *k* invertible mappings called coupling blocks with $f:=f_{1},\ldots ,f_{j},\ldots ,f_{k}$ that predicts $\hat{x}=f\left(y,z;\theta \right)$. The coupling blocks are learnable neural networks, i.e., scaling *s* and translation *t*, such that these functions need not be invertible and can be represented by any neural network [58]. The coupling block takes the input and splits it into two parts, which are transformed by *s* and *t* networks alternatively. The transformed parts are subsequently concatenated to produce the block’s output. The architecture allows for easy recovery of the block’s input from its output in the inverse direction, with minor architectural modifications ensuring invertibility. We follow [59] to perform a learned invertible 1×1 convolution after every coupling block to reverse the ordering of the features, thereby ensuring each feature undergoes the transformation. Hence, the function *f* is a bijective mapping between $\left[y,z\right]$, and x, leading to its invertibility, which help it to associate the conductivity x with unique pairs $\left[y,z\right]$ of magnetic flux density y and latent space z. We incorporate vector z to address the information loss in the forward process, i.e., x→y and to capture the variance in mapping the inverse process, i.e., y→x.

### 4.2. INN Training and Testing Procedure

The algorithm for the training and testing of our proposed INN framework is shown in Algorithm 1. Given that INN as an invertible function *f*, its optimization via training explicitly calculates the inverse process, i.e., $\hat{x}=f\left(y,z;\theta \right)$ where θ are the INN parameters. We define the density of the latent variable $p\left(z\right)$ as the multivariate standard Gaussian distribution. The desired posterior distribution $p\left(x|y\right)$ can now be represented by the deterministic function *f* that pushes the known Gaussian prior distribution $p\left(z\right)$ to the x-space, conditioned on y. Note that the forward mapping $x\to \left[y,z\right]$ through function $f^{-1}$, and the inverse mapping $\left[y,z\right]\to x$ through function *f*, are both differentiable and efficiently computable for posterior probabilities. Therefore, we approximate the conditional probability $p\left(x|y\right)$ by the inverse process of our tractable INN model $f\left(y,z;\theta \right)$, which uses the training data ${\left\{\left(x_{i},y_{i}\right)\right\}}_{i=1}^{T}$ with *T* samples from the forward simulation, as discussed in Section 3. Hence, the objective is to deduce the high-dimensional conductivity distribution x, from a sparse set of magnetic flux density measurements y. Even though our INN can be trained in both directions with losses $L_{x}$, $L_{y}$, and $L_{z}$ for variables x, y, z, respectively, as performed in [10], we are only interested in reconstructing the conductivity variable x, i.e., the inverse process. Given the training batch size as *W*, the loss $L_{x}$ minimizes the reconstruction error between the ground truth and predictions during training as follows:(2)$$L_{x}\left(\theta \right)\;=\;\;{\left(\frac{1}{W}\sum_{i\;=\;1}^{W}{\left|x_{i\;}-\;f\left(y_{i},z_{i},\theta \right)\right|}^{2}\right)}^{\frac{1}{2}}{\;\;\;\;with\;objective\;\;\;\;\;\;\theta }^{*}\;=\;\underset{\theta }{\mathrm{argmin}}\;L_{x}\left(\theta \right)$$
**Algorithm 1:** Training and testing scheme of the invertible neural network
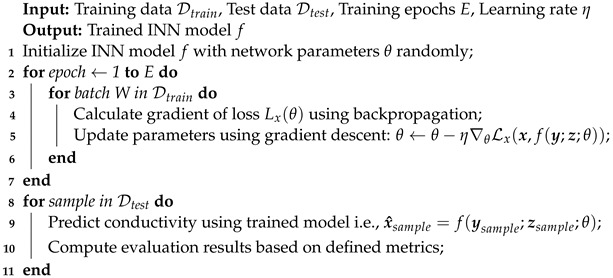


### 4.3. Random Error Diffusion

The ground truth conductivity maps consist of binary values, $x_{sample}$, while the predictions are continuous-valued, ${\hat{x}}_{sample}$. Therefore, it is crucial to define an appropriate metric to assess the performance of the model. In principle, image dithering approaches like Floyd–Steinberg Dithering [69] can be adopted for converting the continuous-valued pixels to binary pixels and then compare its similarity with the ground truth binary map. However, ref. [69] disperses quantization errors into neighboring pixels with pre-defined fractions or a fixed dithering matrix, without adapting to the specific characteristics of the image. Therefore, we developed a novel algorithm named *Random Error Diffusion* [70] (see Algorithm 2) to assess the similarity between the continuous-valued conductivity predictions and the binary-valued ground truth maps. The algorithm utilizes four randomly sampled error fractions from the Dirichlet distribution to diffuse quantization errors in the context of Floyd–Steinberg Dithering. The process is then repeated multiple times to create an ensemble of binary conductivity maps, whose density is estimated. Subsequently, the log-likelihood of the ground truth binary map is estimated with respect to the computed density.
**Algorithm 2:** Random error diffusion
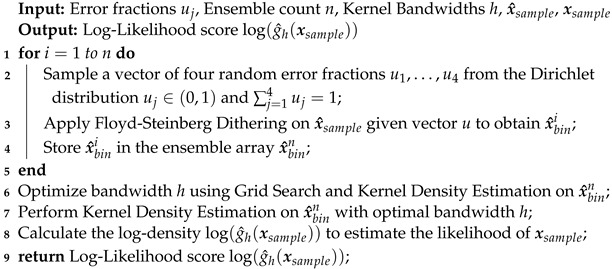


#### Algorithm

To initiate the algorithm, four random error fractions, denoted as $u_{1},\ldots ,u_{4}$, are sampled from the Dirichlet distribution. Each fraction is a real number within the interval (0,1), and their sum is constrained to equal 1. Subsequently, these random error fractions are utilized to diffuse the quantization error to the neighboring pixels in order to obtain the binary conductivity map. This process is repeated *n* times for resampling the four error fractions, which is used to produce an ensemble of *n* binary conductivity maps ${\hat{x}}_{bin}^{n}$, for each continuous valued conductivity prediction ${\hat{x}}_{sample}$. We subsequently perform Kernel Density Estimation (KDE) on the ensembles ${\hat{x}}_{bin}^{n}$ for each conductivity prediction ${\hat{x}}_{sample}$ to obtain the density estimate ${\hat{g}}_{h}$, parameterized by the kernel bandwidth *h*. Finally, the log-likelihood $\log({\hat{g}}_{h}\left(x_{sample}\right))$ of the ground truth binary map $x_{sample}$ is computed from the density estimate ${\hat{g}}_{h}$.

### 4.4. Bias and Deviation

To comprehensively analyze the robustness of the INN and other evaluated models for reconstructing the conductivity distribution amid sensor noise, we introduce two additional evaluation metrics, namely the Bias and Deviation maps. The motivation behind formulating these metrics lies in the observation that the reconstructed conductivity from different evaluated models, as shown in Figure 4, do not reveal the model’s true robustness to noise. Therefore, a noise vector $\delta_{sample}\in R^{M}$ was sampled γ times from the uniform distribution in a pre-defined range. Subsequently, this sampled noise vector $\delta_{sample}$ was added to the magnetic flux density measurements from the validation set $y_{sample}$. The models studied in this work were then utilized along with the noisy magnetic flux density $\left(y_{sample}+\delta_{sample}\right)$ to reconstruct γ conductivity maps, ${\hat{x}}_{sample}$.

*Bias:* Our first metric, denoted as Bias, is computed by first taking the per-pixel average of the γ conductivity maps. Then, the conductivity map predicted from the evaluated model when the sensor readings had no addition of noise is then subtracted from the averaged conductivity map. This results in the computation of the bias map given as:(3)$$Bias\left(p,q\right)=\{\frac{1}{\gamma }\sum_{i=1}^{\gamma }{\hat{x}}_{sample}^{i}\left(p,q\right)\}-{\hat{x}}_{sample}^{0}\left(p,q\right)$$
where Bias(p,q) is the bias at pixel (p,q), γ is the number of iterations, ${\hat{x}}_{sample}^{i}\left(p,q\right)$ is the predicted conductivity at pixel (p,q) in the *i*-th iteration, ${\hat{x}}_{sample}^{0}\left(p,q\right)$ is the predicted conductivity at pixel (p,q) when no noise is added in $y_{sample}$. Thus, the bias map visualizes the model’s tendency to deviate from accurate predictions under different noise conditions.

*Deviation:* We utilized the γ conductivity maps to compute per-pixel standard deviation values, resulting in the deviation map formulated as follows:(4)$$Deviation\left(p,q\right)=\sqrt{\frac{1}{\gamma }\sum_{i=1}^{\gamma }{\left({\hat{x}}_{sample}^{i}\left(p,q\right)-{\bar{x}}_{sample}\left(p,q\right)\right)}^{2}}$$
where Deviation(p,q) is the deviation at pixel (p,q), and ${\bar{x}}_{sample}\left(p,q\right)$ is the average predicted conductivity at pixel (p,q) across all γ iterations. Hence, the per-pixel deviation map estimates the variability in the model’s conductivity predictions across multiple instances of sensor noise. It also elucidates the model’s sensitivity to noise in sensor readings. Together, the bias and deviation maps offer an effective way to analyze the specific strengths and weaknesses of a model to solve the inverse problem, enabling a deeper understanding of the model’s behavior under realistic noisy conditions.

#### Peak Signal-to-Noise Ratio (PSNR)

In our future experimental setup, a uniformly distributed noise may be present in the sensor readings. Our previous study [71] showed that, generally, up to ±10nT noise is observed in similar settings. Therefore, we introduced uniform noise $\delta_{sample}$ within the range of ±1nT, 3nT, 5nT, 10nT, 50nT, 100nT, 500nT, and 1μT. We also evaluated our models on higher noise levels in order to analyze its robustness under atypical sensor anomalies. These noise levels were sampled γ times and was added to the validation set of magnetic flux density measurements, as discussed in Section 4.4. The distance of the sensors from the liquid metal was fixed at d=25 mm with M=50 sensors. To quantify the amount of noise $\delta_{sample}$ added to the magnetic flux density measurements $y_{sample}$ of the validation set, we computed the Peak Signal-to-Noise Ratio (PSNR), expressed in decibels (dB). PSNR measures the logarithmic ratio between the maximum power of the noise-free magnetic flux density measurement, $y_{sample}$ and the mean of the squared noise $\delta_{sample}$ as:(5)$$\mathrm{PSNR}=20\cdot \log_{10}\left(\mathrm{Max}\left(y_{sample}\right)\right)-10\cdot \log_{10}\left(\mathrm{Mean}\left(\delta_{sample}^{2}\right)\right)$$

The PSNR metric quantifies the relationship between the maximum possible signal power and the power of the noise in the signal. A higher PSNR value in this context implies better signal quality, indicating a reduced level of noise or distortion in the magnetic sensor readings. Table 1 presents the average PSNR scores obtained from samples within the validation set of magnetic sensor data. Notably, the noise level up to ±50 nT already results in a low PSNR score. Therefore, the insights from Table 1 prompt further study to visually and quantitatively assess the robustness of the INN model relative to other approaches when reconstructing the conductivity distribution under low PSNR settings.

## 5. Experiments and Results

In this section, we discuss our experimental setup and the obtained results. In Section 5.1, we explain the standardization of the training and test data. Section 5.2 details the meta-parameters defined for training the INN. Finally, we report qualitative results in Section 5.4 and quantitative results in Section 5.5.

### 5.1. Data Standardization

To create distinct training and validation sets, we shuffled the simulated geometries and allocated 80% of the 10,000 geometries for training and 20% for validation. Additionally, we conducted data standardization to facilitate the model’s learning process and enhance convergence efficiency. Standardizing the data ensures that all features share a similar scale, promoting faster convergence, numerical stability, and generalizability. Given the distinct units of measurement for magnetic flux density and conductivity distribution, standardization becomes particularly essential in our case. We specifically employ Z-score normalization as our standardization method, transforming the simulation data to have a per-feature mean value of 0 and a standard deviation of 1. We perform the standardization procedure separately for the magnetic flux density data and binary conductivity distribution.

### 5.2. INN Hyperparameters

The INN model underwent training on four NVIDIA A100 GPUs, utilizing Python 3.8.6 and PyTorch 1.9.0. We fixed the training meta-parameters such as the batch size at 100, optimizer as Adam with a learning rate of $1\times 10^{-4}$, the exponential decay rate for the first and second moment as 0.8 and 0.9, respectively, epsilon score at $1\times 10^{-6}$, and weight decay at $2\times 10^{-5}$. Concerning the INN architecture, we maintained three fully connected layers in *s* and *t* networks for each coupling block. Each layer has 128 neurons and tanh activation function after the first and second layers, whereas there is no activation function in the output layer of the *s* and *t* networks. We studied the effect of the number of coupling blocks for validation loss convergence in Section 5.5.1.

### 5.3. Evaluated Methods

We implemented two distinct coupling block architectures, drawing inspiration from RealNVP [58] and Glow [59] as the backbone of our INN model. Each of these INN models was trained with the loss function described in Equation (Equation 2). We also trained the Glow-based INN model with the Mean Squared Error (MSE) as the objective function such that $L_{x}\left(\theta \right)\;=\frac{1}{W}\sum_{i\;=\;1}^{W}{\left|x_{i\;}-\;f\left(y_{i},z_{i},\theta \right)\right|}^{2}$. The purpose was to assess its performance in terms of reconstructing the conductivity distribution. In addition, we explored three alternative approaches to address the inverse problem at hand, Tikhonov, Elastic Net, and Convolutional Neural Network (CNN). The models Tikhonov and Elastic Net hinge on fitting a linear model regulated by a penalty term. The Tikhonov approach applies an $L_{2}$-Norm penalty on the parameters of the linear model for regularization, while Elastic Net regularization employs a combination of $L_{1}$-Norm and $L_{2}$-Norm penalties on the model parameters. The weights of the regularization term for the Tikhonov and Elastic Net approaches were determined through cross-validation on the training set. To further diversify our evaluation, we introduced a CNN model designed for reconstructing the conductivity distribution. The loss function for the CNN was formulated similarly to Equation (Equation 2). For training the CNN model, we transformed the 100 sensor input data into a 10×10 dimensional input, while the 510 conductivity points were transformed into a 34×15 output 2D map. Further architectural details of the developed CNN model are provided in Table 2. In this paper, we will refer to the six models as INN–Glow, INN–RealNVP, INN–Glow (MSE), Tikhonov, Elastic Net, and CNN as needed.

### 5.4. Qualitative Results

In this section, we present a comprehensive visual comparison of the reconstructed conductivity distribution from several evaluated models. We also report the results of the parameter studies, and discuss the bias and deviation maps obtained from the INN–Glow and Tikhonov model under noisy sensor measurements.

#### 5.4.1. Prediction of the Conductivity Maps: A Comparative Study

In Figure 4, we present the results of predicted conductivity maps by the INN–Glow, INN–RealNVP, Tikhonov, Elastic Net, and the CNN models. These predictions are based on the sensor configuration with d=5 mm and M=100 sensors. It can be observed that both INN–Glow and INN–RealNVP models provides a good approximation of the ground truth conductivity map. The reconstructions reveal pertinent details regarding the locations of non-conducting PMMA cylinder-induced void fraction. The visual outcomes of Tikhonov and Elastic Net regularization exhibit similarities to those of the INN models. In contrast, the CNN model yields a smoother prediction owing to the convolution operation inherent in its architecture. However, the CNN model wrongly predicts the presence of void fraction in regions characterized by high conductivity, as visible in the results of Sample 1. We believe that this occurs due to CNN’s inherent emphasis on learning the local patterns in the image. However, for our specific inverse problem, understanding the global relationship between the bubble distribution and conducting liquid using a fully connected network-based INN acts as a more suitable choice. Furthermore, CNNs are inherently tailored for image processing, while INNs are data agnostic and adaptable to diverse data types. Importantly, INNs are invertible in its design, a property that CNNs lack.

#### 5.4.2. Effect of the Sensor Distance and Number of Sensors

We explored the impact of varying the distance of sensors from the liquid metal, *d*, and the number of sensors, *M* on the quality of the conductivity reconstruction using our INN–Glow model. In this experiment, we trained three separate instances of the INN–Glow model using simulation data, which is based on varying the distance *d* and number of sensors *M*. The first setup is defined with (d=5mm;M=100), the second setup with (d=25mm;M=100), and the third setup as (d=25mm;M=50). Figure 5 present the results obtained from the three example ground truths within the validation set. It shows that the region containing the void fraction is smoother as the distance of the sensors from the liquid metal is increased and the number of sensors is decreased. This outcome can be attributed to the increased difficulty for the model to solve the inverse problem with a lower number of sensors and a greater distance of the sensors from the liquid metal. Nevertheless, the model is effective in reconstructing the arrangement of PMMA cylinder-induced void fraction, also for the third setup with M=50 and d=25 mm.

#### 5.4.3. Robustness to Noise: INN vs. Tikhonov without Noisy Training Data

Based on the method in Section 4.4, we present the results for the reconstruction of the conductivity distribution, bias, and deviation maps after incorporating noise into the validation set of magnetic flux density data. The results are reported after fixing the parameter γ=100 for the INN–Glow model. We also report the results obtained after utilizing the Tikhonov model under the same experimental setup. Note that the training data did not contain noise in the sensor readings.

*Conductivity Maps:* In Figure 6, the left column shows the INN–Glow model’s robustness in reconstructing the conductivity distribution, even with the presence of uniform noise $\delta_{sample}$ up to ±100 nT in the magnetic flux density data. In contrast, the first column of Figure 7 conveys a noteworthy decline in Tikhonov’s performance to reconstruct conductivity distribution, evident even with ±3 nT noise in the sensor data. This discrepancy results from the Tikhonov model’s inherent linearity, making it highly susceptible to noise perturbations. In contrast, the INN–Glow, with its inherent non-linearity is resilient to noise, resulting in visually superior performance compared to Tikhonov.

*Bias and Deviation Maps:* The middle column in Figure 6 and Figure 7 illustrates bias maps for INN–Glow and Tikhonov, respectively. The results show that the Tikhonov model has a high bias, indicating a higher instability in its conductivity predictions when exposed to varying noise within the same noise value range. In contrast, the INN–Glow model exhibits minimal bias and has a high level of robustness for reconstructing conductivity maps with the presence of noise up to ±100nT in the sensor readings. The right column in Figure 6 and Figure 7 shows the deviation maps for INN–Glow and Tikhonov, respectively. The per-pixel standard deviation of the conductivity maps obtained from the Tikhonov model (see color bars of the deviation maps) linearly increases from noise level ±1nT to ±1 μT. On the contrary, the INN–Glow model shows resilience with consistently low per-pixel deviation, that only rises after sensor readings are perturbed with the ±100 nT noise level. These results convey that Tikhonov model, due to its linearity, is markedly more susceptible to noise than the INN–Glow model.

#### 5.4.4. Robustness to Noise: INN vs. Tikhonov with Noisy Training Data

In this section, we compare the results obtained from INN–Glow and Tikhonov models after the noise levels of ±3 nT and ±50 nT were added to the sensor measurements during training. The parameter γ is set at 100, and we show the reconstructed conductivity distribution, bias, and deviation maps at varying level of noise during testing.

*Conductivity Maps:* The left column of Figure 8 and Figure 9 shows the reconstruction of the conductivity maps obtained from the INN–Glow model trained with ±3 nT and ±50 nT noise in the training data, respectively. Additionally, the left column of Figure 10 and Figure 11 shows the reconstruction of the conductivity maps for the Tikhonov model at ±3 nT and ±50 nT noise in the training data, respectively. It is evident that for ±3 nT noise in training data, the INN–Glow model exhibit robustness to predict the void fraction up to ±50 nT noise in the validation example, while the Tikhonov model precisely reconstructs conductivity up to ±10 nT noise in the validation example. However with ±50 nT noise in the training data, the reconstruction of the conductivity distribution from both the Tikhonov and INN–Glow model are robust until ±100 nT noise in the validation example.

*Bias and Deviation Maps:* The middle and right columns of Figure 8 and Figure 9 show the bias and deviation maps obtained from the INN–Glow model at ±3 nT and ±50 nT noise in the training data, respectively, while the middle and right columns in Figure 10 and Figure 11 display the bias and deviation maps for the Tikhonov model. The results for ±50 nT noise in the training data reveals that until ±100 nT noise in the validation example, the Tikhonov model has lower bias and deviation than the INN–Glow model. With the presence of similar noise levels in both training and validation data, a linear model like Tikhonov typically has a low bias while models like INN–Glow can produce higher bias due to their inherent non-linearity. However, both the INN–Glow and Tikhonov models exhibit high bias and deviation at ±500 nT and ±1 μT noise levels in the validation example.

#### 5.4.5. Robustness to Noise: Summary

To summarize, the results from Section 5.4.3 and Section 5.4.4 show that the INN–Glow model performs better than the Tikhonov model if trained without noise and tested with noise in sensor measurements. This finding holds for a large range of noise levels. However, if the noise level is known during model training, the Tikhonov model performs as good as our INN model for reconstructing conductivity maps with lower bias and deviation for the reconstruction. Therefore, for the future experimental setups, if the noise level is not known or if the noise is varying based on the properties of the sensor measurements or further external influences, we can perform INN–Glow training without incorporating noise and then utilize the trained INN–Glow model to precisely reconstruct the conductivity maps in the presence of noise in sensor readings, even if the noise level changes significantly.

#### 5.4.6. Effect of Number of Uniform Noise Samples

We conducted a parameter study to analyze the significance of the number of uniform noise samples γ on the bias and deviation computation for reconstructing the conductivity maps. For this experiment, we fixed the noise level of ±100 nT, and the results are presented in Figure 12, for γ at 10, 100, and 1000 samples. It is apparent that γ has a pronounced effect on the Tikhonov model, reducing bias more significantly compared to the INN–Glow model when γ is higher. Furthermore, there is less effect of varying γ on the deviation maps for both models. The results affirm that an increase in the γ value tends to reduce bias, but fixing a very high value of γ may result in substantial computational requirements.

#### 5.4.7. Random Sampling from Latent Space

We analyzed the influence of random sampling from the normally distributed latent space $z_{sample}$ on the INN model’s robustness for reconstructing the conductivity distribution. We sampled the latent space $z_{sample}$ multiple times, and alongside the magnetic flux density measurements $y_{sample}$, we passed $\left[z_{sample},y_{sample}\right]$ to the INN–Glow model for the reconstruction of the conductivity distribution. This sampling procedure was repeated 100 times, and we computed bias and deviation maps following the similar protocol established in previous experiments. The results, illustrated in Figure 13 for the example validation ground truth, show that random sampling from the latent space $z_{sample}$ causes minimal bias and deviation on the quality of the reconstruction of the conductivity distribution. This observation is evident in the three examples of the predicted conductivity distributions as shown in Figure 13d–f from three different latent $z_{sample}$ vectors and low bias and deviation scores as shown in Figure 13b,c, respectively.

### 5.5. Quantitative Results

In this section, we provide quantitative results for a thorough evaluation of the proposed models for solving the inverse problem. We discuss key performance metrics, such as the random error diffusion, average bias, and average deviation scores, to assess each of the evaluated model’s qualities of the reconstructing conductivity distribution.

#### 5.5.1. Effect of Number of Coupling Blocks on Validation Loss

Figure 14 illustrates the impact of the number of coupling blocks *k* of the INN–Glow model on the convergence of validation loss. We stop the model training when the validation loss begins to increase. The loss curves reveal that a single coupling block leads to underfitting, while a higher number of blocks may result in overfitting without the stoppage of the training iterations. Figure 14a–c show that the configuration d=25 mm and M=100 has higher validation loss compared to the setup with d=5 mm and M=100 due to reduced information in magnetic flux density measurements with a greater sensor distance from the liquid metal. Additionally, the configuration with d=25 mm and M=50 sensors further degrades information, leading to much higher loss while solving the inverse problem. Despite the inferior loss convergence, Figure 5 demonstrated the INN–Glow model’s ability to learn the location of void fraction for the configuration with d=25 mm and M=50 sensors. Notably, increasing the number of coupling blocks beyond k=3 does not substantially reduce validation loss, as the loss scores at the last epoch before the training stoppage as shown in Figure 14d reveals.

#### 5.5.2. Random Error Diffusion

We compared the results obtained from the random error diffusion metric presented in Section 4.3 for the six different models to solve the inverse problem. The results in Figure 15 show the log-likelihood distribution of all the 2000 validation ground truth samples for varying counts of binary ensembles *n*. It can be seen that the log-likelihood scores are centered near zero irrespective of the model, and the ensemble count *n*. This outcome can also be verified by the averaged log-likelihood scores in Table 3. Figure 15 and Table 3 show that for both n=100 and 1000, the INN–Glow and INN–RealNVP models perform better than the linear models, i.e., Tikhonov and Elastic Net as well as INN-Glow (MSE) as they achieve higher average log-likelihoods. However, the CNN model has a higher log-likelihood score than all other evaluated models. Due to the convolution operation, the CNN model predicts blurred images. The blurring obscures fine details and feature edges and makes the image appear more uniform and less detailed, similar to a binary map. Hence, random error diffusion estimates higher likelihoods that these blurred images are being sampled from the density of binary ensembles.

#### 5.5.3. Bias and Deviation

Table 4 presents the quantitative results related to bias and deviation maps for the INN–Glow and Tikhonov models. To compute the deviation score, we took the average of the deviation maps across all the 2000 validation samples for different noise levels. Additionally, for computing both bias (min) and bias (max), we determine the minimum and maximum bias scores from all 2000 validation bias maps. The results in Table 4 indicate that the INN–Glow model consistently exhibits much lower deviation and bias scores compared to the Tikhonov model. This underscores the INN–Glow model’s stability and robustness in reconstructing conductivity maps in the presence of noise in sensor readings during testing, when there is no noise during training. Conversely, the Tikhonov model is less reliable, especially when subjected to noise beyond ±10 nT in sensor readings.

#### 5.5.4. Number of Uniform Noise Samples

Table 5 displays the average deviation and bias scores for varying values of γ. The results indicate that a higher number of noise samplings lead to reduced bias, but a minimal change in the deviation scores, which is consistent with our findings in Figure 12. Notably, the Tikhonov model shows a significant reduction in bias scores, suggesting its sensitivity to the choice of γ. Similarly, the INN–Glow model’s sensitivity to γ is evident, although the impact is less pronounced given its already low bias scores. Given the results in Table 5, we fixed γ=100 for our experiments as this value provides a good balance between the computational requirements and the model’s performance.

## 6. Conclusions

In this study, we introduced Invertible Neural Networks (INNs) for the reconstruction of conductivity distribution from external magnetic field measurements under simulation conditions similar to those encountered in a water electrolyzer. Our results highlight the robustness of the INN model, showcasing its ability to learn conductivity distributions in the face of the inherently ill-posed nature of the problem and the presence of noise in magnetic flux density measurements. In contrast, linear models like Tikhonov exhibit high susceptibility to noise, due to which the reconstructions from such models are unreliable beyond a certain noise level in sensor readings of the test data, especially when the model is fitted with sensor data containing no noise. The extensive evaluation, involving bias, deviation, and random error diffusion metrics, underscore the superior performance of the INN model in approximating ground truth conductivity maps compared to the Tikhonov model. Additionally, our findings suggest that INNs can efficiently reconstruct conductivity maps even with a limited number of sensors, positioned at distances exceeding 20 mm from the conducting plate. Our INN model’s real-time prediction capabilities have practical applications, especially in estimating the void fraction distributions within actual electrolysis cells. This positions INNs as a promising model for localizing and estimating bubble void fraction locations in current-conducting liquids. In the future, we will focus on evaluating INNs for bubble and void fraction detection within experimental electrolysis setups and also test the findings from this work in other inverse problems of applied physics.

## Figures and Tables

**Figure 1 sensors-24-01213-f001:**
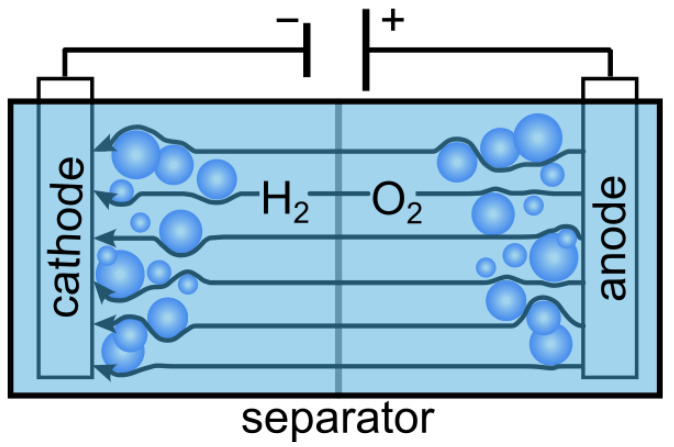
The illustration provides a visual representation of an electrolysis cell, elucidating the notable occurrence of bubble formation concentrated specifically at the electrode reaction sites.

**Figure 2 sensors-24-01213-f002:**
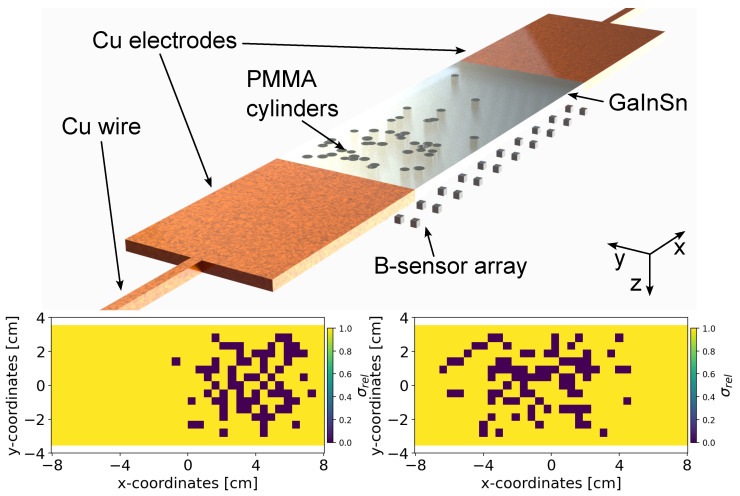
The top figure shows the Proof-of-Concept (POC) model that contains a channel filled with liquid GaInSn with Poly-Methyl Methacrylate (PMMA) cylinders normally distributed along the *x*-axis and randomly distributed along the *y*-axis in the channel. The top figure also shows the Cu electrodes with wire to apply electric current to the plate, and the magnetic sensors on the bottom. The two bottom figures show examples of the binarized conductivity distribution of liquid metal-containing region in the xyz cartesian plane. The dark pixels resemble low conductivity meaning the presence of void fraction clusters.

**Figure 3 sensors-24-01213-f003:**
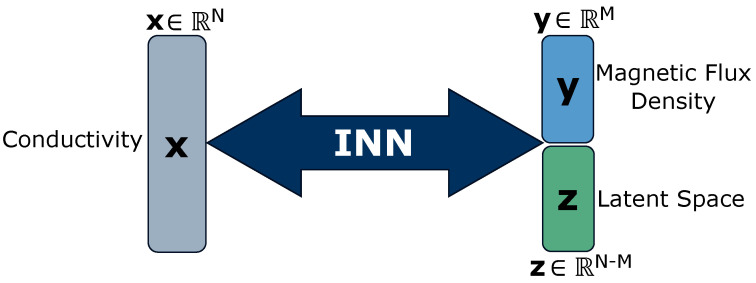
An overview of our Invertible Neural Network (INN) architecture. The conductivity map x is positioned on the left side of the network. The INN architecture contains *k* coupling blocks. On the right side of the network are variables y and z, i.e., magnetic flux density and latent space, respectively. The INN is trainable in both directions, as shown with the bi-directional arrows in the figure.

**Figure 4 sensors-24-01213-f004:**
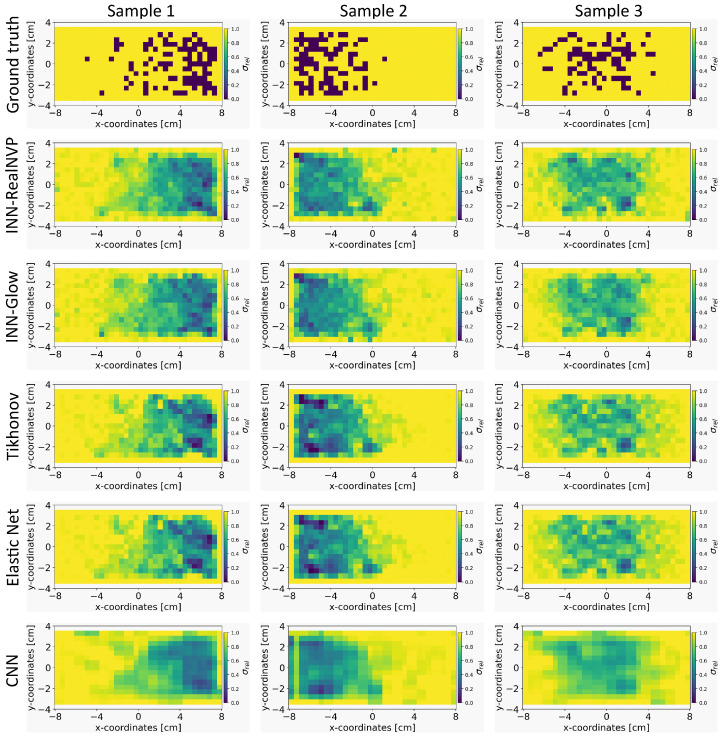
Visual comparison of the quality of the reconstruction of the conductivity distribution $x_{sample}$ from example ground truths of the validation set on the evaluated models. We used the simulation configuration of d=5 mm with M=100 sensors.

**Figure 5 sensors-24-01213-f005:**
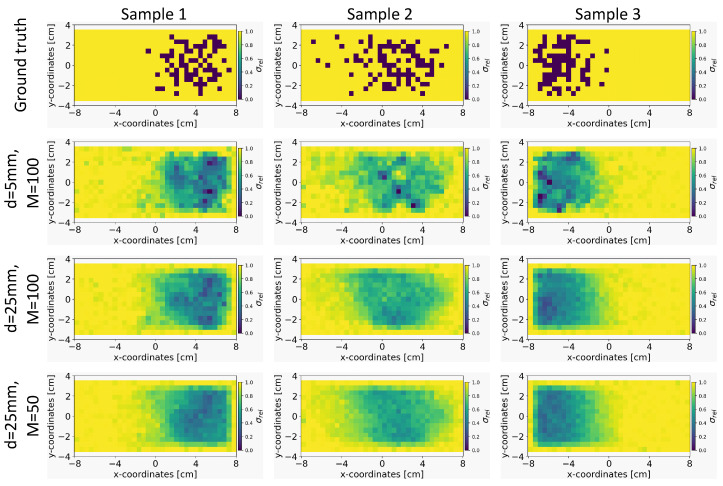
Comparison of the reconstruction quality of the conductivity distribution for the Invertible Neural Network (INN)–Glow model after varying the simulation parameters, such as distance from the liquid metal *d* and the number of sensors *M*.

**Figure 6 sensors-24-01213-f006:**
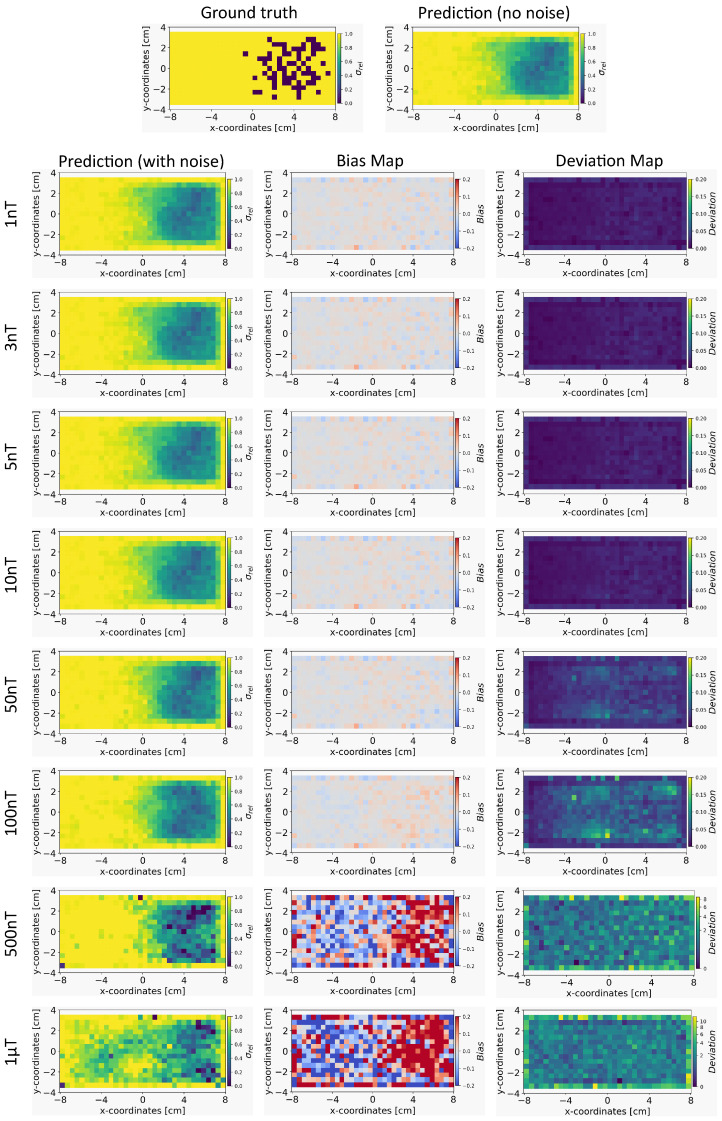
The figure shows the reconstruction of the conductivity maps (left column) and the corresponding bias (middle column) and deviation maps (right column) obtained from the INN–Glow model at different noise levels with d=25 mm, M=50 sensors, and γ=100. The INN–Glow model is trained with magnetic flux density measurements that have no noise in the sensor readings.

**Figure 7 sensors-24-01213-f007:**
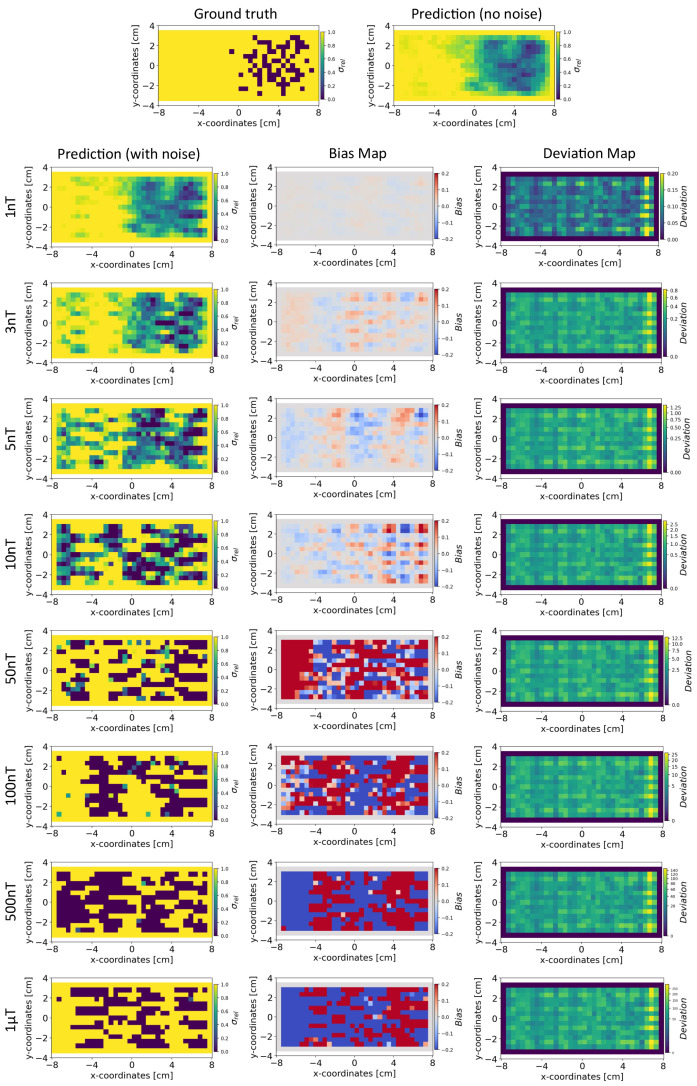
The figure shows the reconstruction of the conductivity maps (left column) and the corresponding bias (middle column) and deviation maps (right column) obtained from the Tikhonov model at different noise levels with d=25 mm, M=50 sensors, and γ=100. The Tikhonov model is fitted with magnetic flux density measurements that have no noise in the sensor readings.

**Figure 8 sensors-24-01213-f008:**
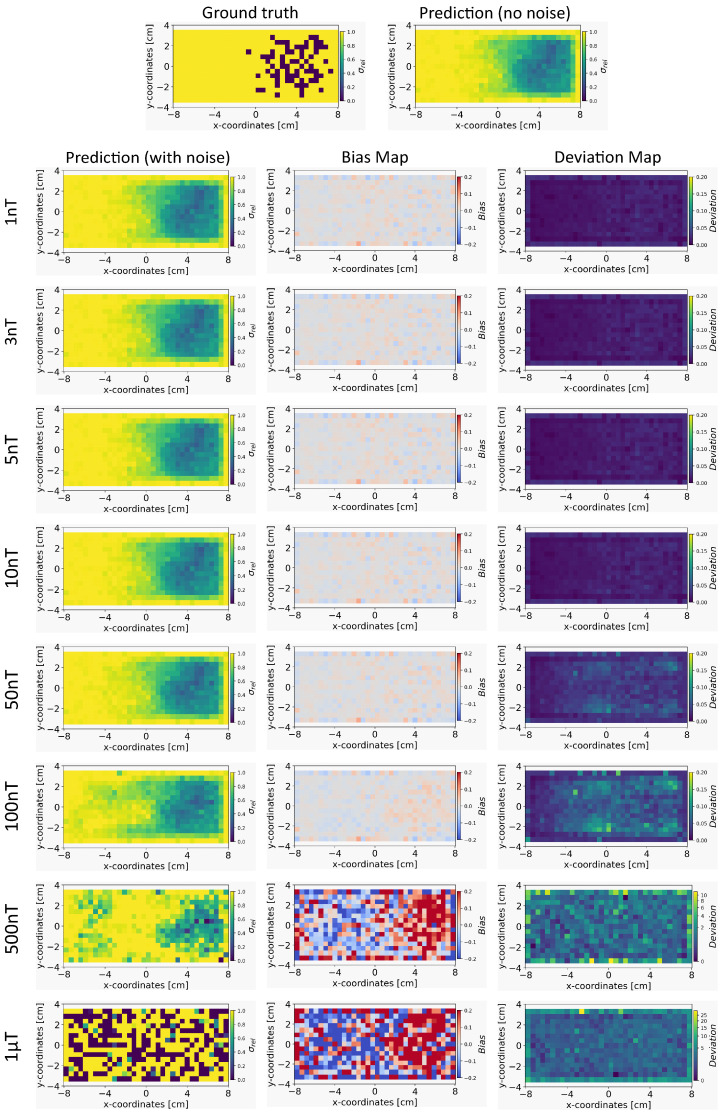
The figure shows the reconstruction of the conductivity maps (left column) and the corresponding bias (middle column) and deviation maps (right column) obtained from the INN–Glow model at different noise levels with d=25 mm, M=50 sensors, and γ=100. The INN–Glow model is trained with magnetic flux density measurements that have ±3 nT uniformly distributed noise in the sensor readings.

**Figure 9 sensors-24-01213-f009:**
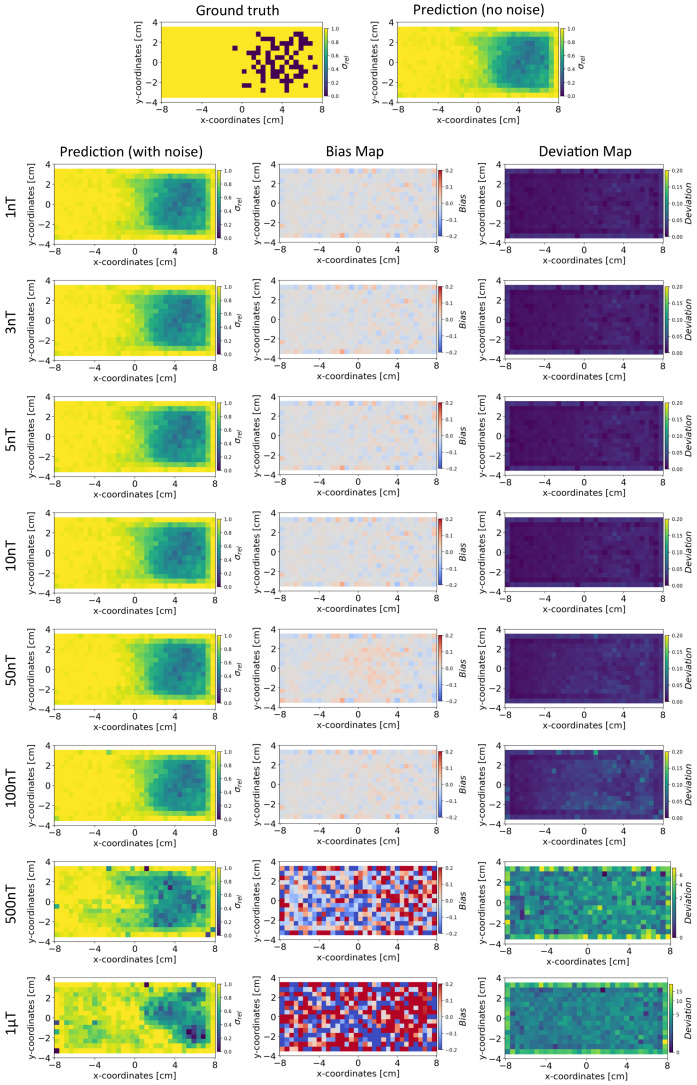
Reconstruction of the conductivity maps (left column) and the corresponding bias (middle column) and deviation maps (right column) obtained from the Invertible Neural Network (INN)–Glow model at different noise levels with d=25 mm, M=50 sensors, and γ=100. The INN–Glow model is trained with magnetic flux density measurements that have ±50 nT uniformly distributed noise in the sensor readings.

**Figure 10 sensors-24-01213-f010:**
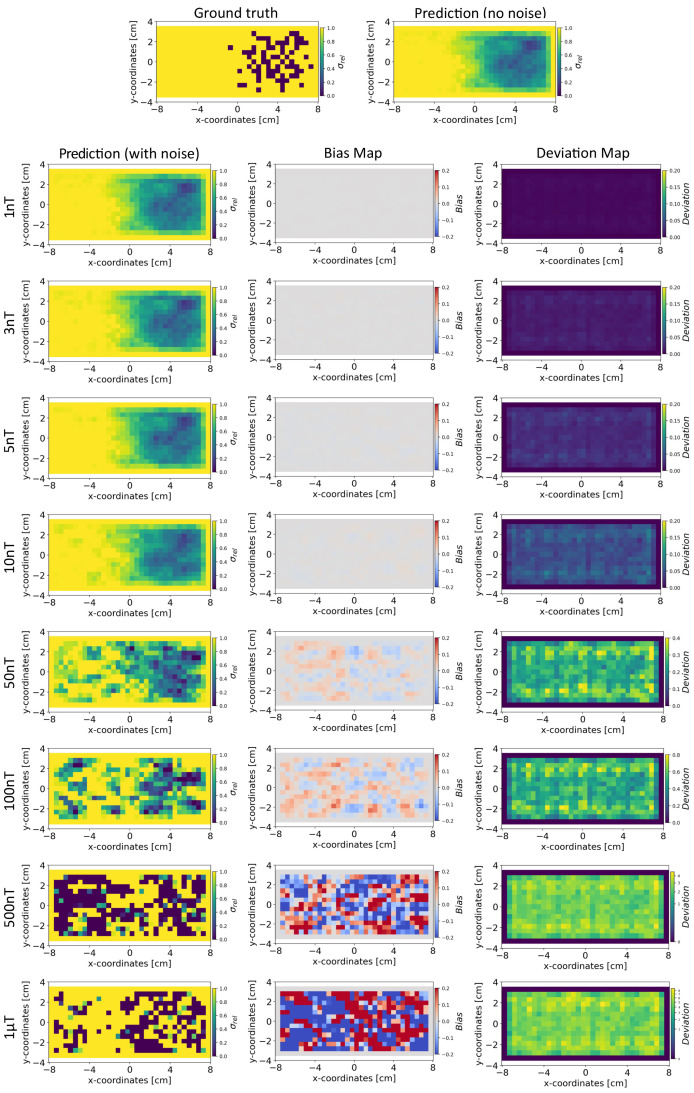
Reconstruction of the conductivity maps (left column) and the corresponding bias (middle column) and deviation maps (right column) obtained from the Tikhonov model at different noise levels with d=25 mm, M=50 sensors, and γ=100. The Tikhonov model is fitted with magnetic flux density measurements that have ±3 nT uniformly distributed noise in the sensor readings.

**Figure 11 sensors-24-01213-f011:**
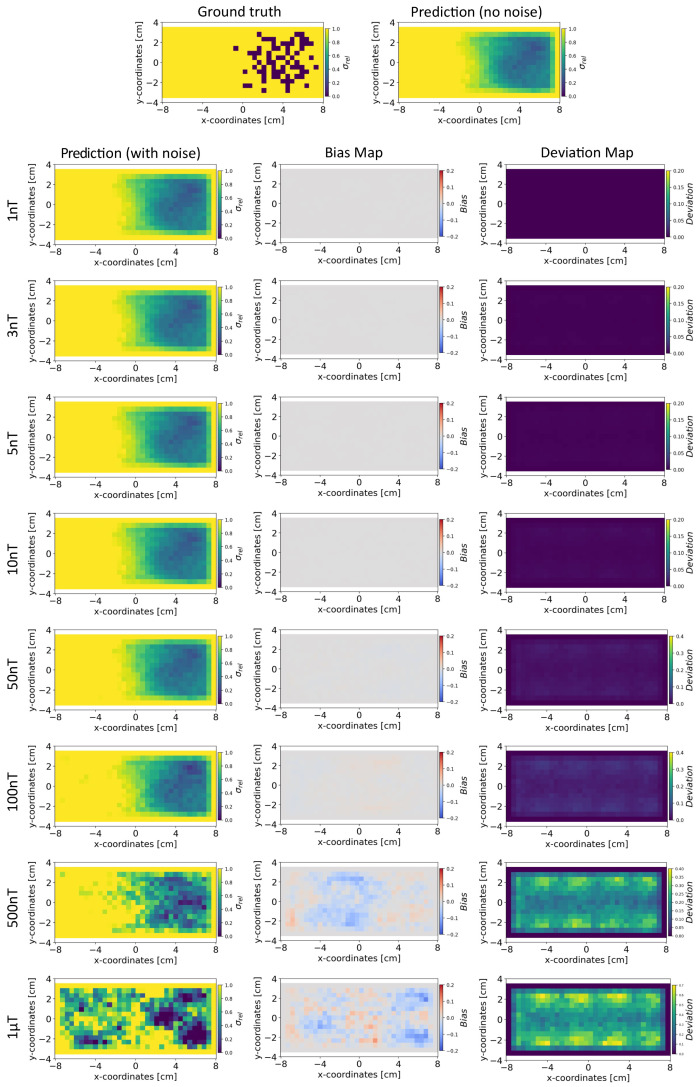
Reconstruction of the conductivity maps (left column) and the corresponding bias (middle column) and deviation maps (right column) obtained from the Tikhonov model at different noise levels with d=25 mm, M=50 sensors, and γ=100. The Tikhonov model is fitted with magnetic flux density measurements with ±50 nT uniformly distributed noise in the sensor readings.

**Figure 12 sensors-24-01213-f012:**
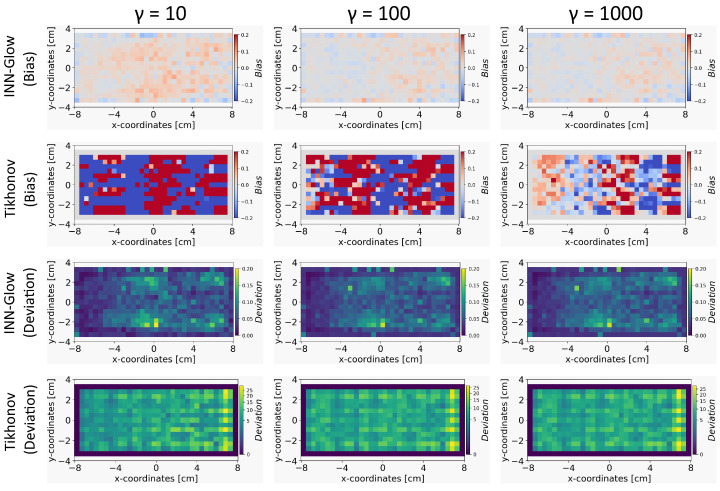
The figure shows the bias and deviation maps for Invertible Neural Network (INN)–Glow and Tikhonov models after varying the parameter γ. The results are for the validation ground truth example in Figure 6. We used the noise range ±100 nT in the sensor data, and no noise was added during the training.

**Figure 13 sensors-24-01213-f013:**
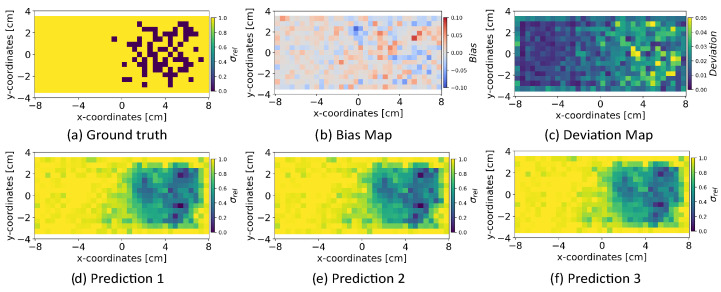
The figure shows the results after random sampling from the latent space $z_{sample}$ of the INN–Glow model. The bottom row shows examples of the reconstructed conductivity distribution after varying $z_{sample}$. The model is trained with the magnetic flux density measurements consisting of no noise in training and validation data and simulation parameters are d=5 mm, M=100 sensors.

**Figure 14 sensors-24-01213-f014:**
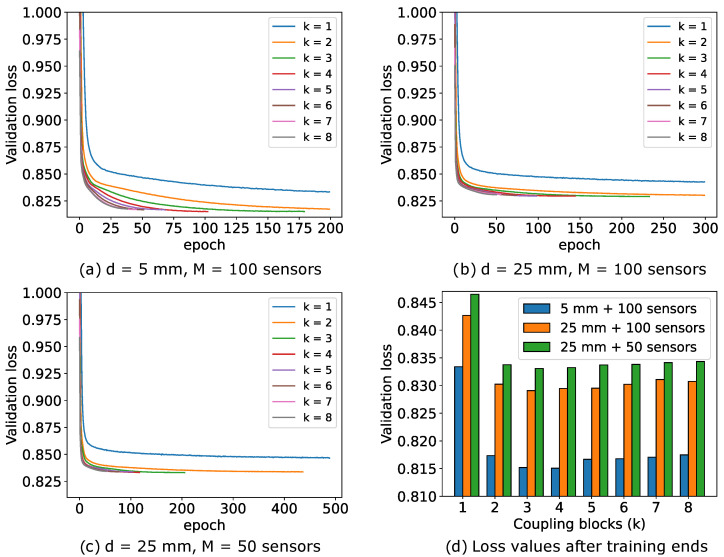
The validation loss curves of multiple instances of the Invertible Neural Network (INN)–Glow models with varying numbers of coupling blocks, denoted as *k*, and under varying values of the parameters *d* and *M*.

**Figure 15 sensors-24-01213-f015:**
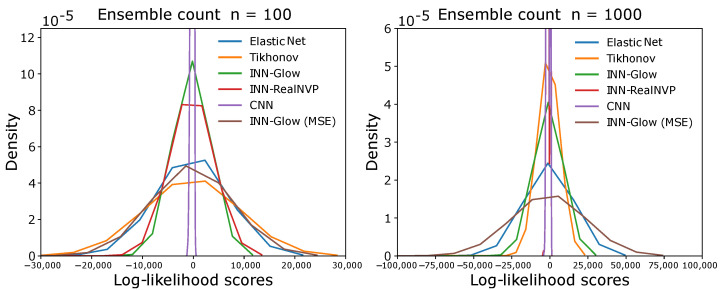
The figure shows the distribution of log-likelihood scores for all the validation ground truth conductivity samples with respect to the probability distribution of binary ensemble maps via random error diffusion. The left and right figures are for ensemble counts of 100 and 1000, respectively.

**Table 1 sensors-24-01213-t001:** Average peak signal-to-noise ratio for the validation set of the ground truth magnetic flux density data. The distance of the sensors from the liquid metal is d=25 mm with M=50 sensors.

Noise	1nT	3nT	5nT	10nT	50nT	100nT	500nT	1μT
PSNR (dB)	56.46	46.93	42.51	36.48	22.50	16.49	2.48	−3.52

**Table 2 sensors-24-01213-t002:** Architecture of the developed Convolutional Neural Network (CNN) for the simulation configuration of M=100 sensors and the sensor distance of d=5 mm from the liquid metal.

Layer Type	Number of Filters	Feature Size	Kernel Size	Strides
Image Input Layer		10 × 10 × 1		
1st convolution layer	32	10 × 10 × 32	[3, 3]	[1, 1]
ReLU Layer				
2nd convolution layer	64	10 × 10 × 64	[3, 3]	[1, 1]
ReLU Layer				
3rd convolution layer	128	5 × 5 × 128	[4, 4]	[2, 2]
ReLU Layer				
4th convolution layer	128	5 × 5 × 128	[3, 3]	[1, 1]
ReLU Layer				
5th convolution layer	64	5 × 5 × 64	[3, 3]	[1, 1]
Nearest Neighbor Upsampling		10 × 10 × 64		
6th convolution layer	32	10 × 10 × 32	[3, 3]	[1, 1]
Nearest Neighbor Upsampling		20 × 20 × 32		
7th convolution layer	1	20 × 20 × 1	[3, 3]	[1, 1]
Nearest Neighbor Interpolation		34 × 15 × 1		

**Table 3 sensors-24-01213-t003:** Averaged log-likelihood scores based on random error diffusion from the validation ground truth samples. The simulation parameters are fixed at d=5 mm and M=100 sensors.

Model	INN-Glow	INN-Glow (MSE)	INN-RealNVP	Tikhonov	Elastic Net	CNN
n=100	−662.64	−1658.68	−910.38	−1907.59	−1291.45	−258.37
n=1000	−2000.93	−5199.36	−1040.56	−2241.14	−3042.36	−975.01

**Table 4 sensors-24-01213-t004:** Average bias and deviation scores with respect to validation ground truth at *d* = 25 mm, M=50 sensors, and γ = 100 for different noise levels. The models being used are INN–Glow and Tikhonov, and during training, the data does not contain any noise in the sensor readings.

Metric	Model	1 nT	3 nT	5 nT	10 nT	50 nT	100 nT	500 nT	1 μT
Deviation	INN-Glow	0.015	0.016	0.016	0.018	0.043	0.073	1.648	3.144
	Tikhonov	0.069	0.206	0.344	0.687	3.437	6.869	34.337	68.679
Bias (min)	INN-Glow	−0.160	−0.160	−0.160	−0.161	−0.173	−0.233	−3.778	−9.937
	Tikhonov	−0.09	−0.315	−0.483	−1.185	−5.050	−10.157	−57.615	−107.509
Bias (max)	INN-Glow	0.227	0.227	0.227	0.227	0.229	0.273	5.280	10.670
	Tikhonov	0.099	0.290	0.485	1.272	5.123	10.093	53.572	101.363

**Table 5 sensors-24-01213-t005:** Average bias and deviation scores with respect to all the validation ground truth geometries at d=25 mm, M=50 sensors, noise level fixed at ±100 nT, and varying γ. The results are from the INN–Glow model, and the training data are without the presence of noise in the sensor readings.

Metric	Model	γ = 10	γ = 100	γ = 1000
Deviation	INN-Glow	0.070	0.073	0.073
	Tikhonov	6.93	6.87	6.92
Bias (min)	INN-Glow	−0.897	−0.233	−0.183
	Tikhonov	−33.274	−10.157	−3.027
Bias (max)	INN-Glow	0.856	0.273	0.316
	Tikhonov	28.074	10.093	2.949

## Data Availability

Data can be provided upon the request to the authors.

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
