# Peer review of "Robust Reconstruction of the Void Fraction from Noisy Magnetic Flux Density Using Invertible Neural Networks"

_sensors, 2024, doi:10.3390/s24041213_

Round 1
Reviewer 1 Report
Comments and Suggestions for Authors
The manuscript describes the reconstruction of conductivity distribution of a simulation condition via external magnetic field measurement. Invertible Neural Networks(INNs) were employed to solve the inverse problem of Bio-Savart’s Law. INN model is robust, efficient, and reliable for the reconstruction of the binarized conductivity distribution of Poly-methyl methacrylate (PPMA) cylinders surrounded by liquid GaInSn. This INN model is promising for estimating the void fraction of bubbles in an electrolysis cell. This manuscript is well-organized and containing interesting details. Prior publication, I suggest revising the manuscript to address at least following concern:
1. On page6, Section 3.1-line 238, is the connection length between the anode and cathode are significant in this model simulation? Is 50 cm the most appropriate value for this simulation?
2. On page7, Section 3.4-line 284, the calculation time of 2.5 min. and the mesh transformation of 3.5 min are quite significant for the model. Is the reconstruction from the model represent the real-time measurement of a sample?
3. On page11, Section 4.4.1-line 417, the uniform noises between ± 1nT and 3nT are quite small. If the measured valued by a magnetic sensor are deviated about ± 1nT, what would be contribute to their PSNR?
4. On page23, Section 5.5, for an experiment, the interference of magnetic flux generated form near neighbor sensors can introduced a sort of an error. Is the INN model suitable for solving this issue as well?
Reviewer 2 Report
Comments and Suggestions for Authors
The authors report Invertible Neural Networks (INNs) for the reconstruction of conductivity distribution from external magnetic field measurements under simulation conditions. Actually, this is a good work. Here I have some comments, please consider them.
1. The authors say their previous study shows that up to ± 10nT noise is observed in similar settings. So what the meaning to set the uniform noise much larger than ± 10nT for their experiment? Please explain it.
2.please cite more references to compare their results with others’ work, since almost no citation is given in 4, 5 sections of their main manuscript.
3.please add more references for their work since they can cite more references for one work more than 25 pages.
